# Reward-predictive representations generalize across tasks in reinforcement learning

**Lucas Lehnert**[1,3]*, **Michael L. Littman**[1], **Michael J. Frank**[2,3]

**1** Computer Science Department, Brown University, Providence, RI 02912, USA, **2** Department of Cognitive, Linguistic & Psychological Sciences, Brown University, Providence, RI 02912, USA, **3** Carney Institute for Brain Science, Brown University, Providence, RI 02912, USA

* lucas_lehnert@brown.edu

**Data Availability Statement:** All implementations are available on Github at https://github.com/lucaslehnert/rewardpredictive. All datasets generated by the conducted simulations are publicly available on Docker Hub at https://hub.

## Abstract

In computer science, reinforcement learning is a powerful framework with which artificial agents can learn to maximize their performance for any given Markov decision process (MDP). Advances over the last decade, in combination with deep neural networks, have enjoyed performance advantages over humans in many difficult task settings. However, such frameworks perform far less favorably when evaluated in their ability to generalize or transfer representations across different tasks. Existing algorithms that facilitate transfer typically are limited to cases in which the transition function or the optimal policy is portable to new contexts, but achieving "deep transfer" characteristic of human behavior has been elusive. Such transfer typically requires discovery of abstractions that permit analogical reuse of previously learned representations to superficially distinct tasks. Here, we demonstrate that abstractions that minimize error in predictions of reward outcomes generalize across tasks with different transition and reward functions. Such reward-predictive representations compress the state space of a task into a lower dimensional representation by combining states that are equivalent in terms of both the transition and reward functions. Because only state equivalences are considered, the resulting state representation is not tied to the transition and reward functions themselves and thus generalizes across tasks with different reward and transition functions. These results contrast with those using abstractions that myopically maximize reward in any given MDP and motivate further experiments in humans and animals to investigate if neural and cognitive systems involved in state representation perform abstractions that facilitate such equivalence relations.

## Author summary

Humans are capable of transferring abstract knowledge from one task to another. For example, in a right-hand-drive country, a driver has to use the right arm to operate the shifter. A driver who learned how to drive in a right-hand-drive country can adapt to operating a left-hand-drive car and use the other arm for shifting instead of re-learning how to drive. Despite the fact that both tasks require different coordination of motor skills, both tasks are the same in an abstract sense: In both tasks, a car is operated and

docker.com/r/lucasdocker/rewardpredictive. Please refer to the documentation provided in the Github repository on how to download and open the provided datasets.

**Funding:** Lucas Lehnert was funded in part by NIH T32MH115895 Training program for Interactionist Cognitive Neuroscience (https://www.nih.gov). Michael L. Littman was funded in part by the ONR MURI PERISCOPE project. Michael J. Frank was funded in part by grant NIH MH084840 (https://www.nih.gov). The funders had no role in study design, data collection and analysis, decision to publish, or preparation of the manuscript.

**Competing interests:** The authors have declared that no competing interests exist.

there is the same progression from 1st to 2nd gear and so on. We study distinct algorithms by which a reinforcement learning agent can discover state representations that encode knowledge about a particular task, and evaluate how well they can generalize. Through a sequence of simulation results, we show that state abstractions that minimize errors in prediction about future reward outcomes generalize across tasks, even those that superficially differ in both the goals (rewards) and the transitions from one state to the next. This work motivates biological studies to determine if distinct circuits are adapted to maximize reward vs. to discover useful state representations.

## Introduction

A central question in reinforcement learning (RL) [1] is which representations facilitate re-use of knowledge across different tasks. Existing deep reinforcement learning algorithms, such as the DQN algorithm [2], construct latent representations to find a reward-maximizing policy in tasks with complex visual inputs. While these representations may be useful for abstracting across states in the service of optimal performance in a specific task, this article considers representations that facilitate re-use across different tasks. Humans are adept at such flexible transfer. As a lay example, consider shifting gears in a manual transmission car. In a right-hand-drive country, the steering wheel is on the left side of the car and the right arm is used for shifting, whereas the opposite is the case in a left-hand-drive country. A person who has learned in one scenario can quickly generalize to the other, despite the fact that both tasks require different coordination of motor skills. Both tasks are the same in an abstract sense: In each case, there is a progression from 1st to 2nd gear and so on, which should be coordinated with the clutch pedal and steering, and this structure can be generalized from a left-hand-drive car to a right-hand-drive car [3, 4] and a driver does not have to learn how to drive from scratch.

By treating two different sensory inputs or states as equivalent, an agent can generalize what it has learned from one state to another and speed up learning [4, 5]. Such equivalences can be modeled using *state abstractions* [3, 6], which map states to a compressed latent representation. The usual RL framework considers Markov Decision Processes (MDPs) [7] in which an agent's sole objective is to maximize reward. In contrast, in transfer or lifelong learning, an agent observes a sequence of MDPs and attempts to learn a state abstraction that can be re-used to speed up learning in a previously unseen task. State abstractions can be constructed in different ways, for example by merging states with the same optimal action or Q-values into the same latent or *abstract state*. This article considers two types of state abstractions:

1. *reward-maximizing state abstractions*, which allow an agent to maximize total reward, and

2. *reward-predictive state abstractions*, which allow an agent to predict future reward sequences.

While many different RL transfer algorithms have been proposed (see [8] for a survey), this article demonstrates that, while reward-maximizing state abstractions are useful for compressing states within a given task, they fail to generalize across tasks that differ in reward and transition functions. In contrast, reward-predictive state abstractions can be leveraged to improve generalization even when both transition and reward functions change across tasks. The presented analysis and simulations motivate the design of new RL algorithms that can discover

such state abstractions as well as further experiments to investigate whether neural mechanisms in biological agents facilitate learning of such representations.

Previous work [9] shows that reward-predictive state abstractions can be extracted from the *successor representation* (SR) [10], which predicts the discounted expected frequency of visiting future states given the current state. While re-using a previously learned SR has been shown to speed up learning when reward functions change [11–13], these methods are only suitable if transition functions are shared (e.g., if one is in the same maze but only the location of the goal changes). Further, if the optimal decision-making strategy differs between two tasks, the SR has to be re-learned [14]. In contrast, this article shows that reward-predictive state abstractions afford "zero-shot" transfer across tasks with variations in transition functions, reward functions, and optimal policies and do not have to be adjusted or re-learned for each task. Such "deep transfer" across environments, even in the absence of prior experience with specific transition or reward functions, is predicted by behavioral and neural signatures of human structure learning [4, 15, 16] but not afforded by alternative algorithms that compress the transition function itself directly [17, 18].

To unpack the relative advantages of distinct state abstraction algorithms for generalization, we proceed as follows. In the following section, we begin with a simple illustration of the state-abstraction framework and then present the conceptual utility of reward-predictive state abstractions. Next, we present our first result by examining this advantage quantitatively when a single abstraction is possible for re-use across a range of task settings and assumptions about the number of latent states (Transfer with single state abstractions). Subsequently, we consider a curriculum-learning situation where multiple state abstractions might apply to different MDPs and the agent has to select amongst them when learning a new MDP (Transfer with multiple state abstractions). Extending our simulations to an online learning setting, we show that this advantage is preserved even when the agent has to simultaneously learn the transitions and rewards of the new MDP and perform inference (Learning to transfer multiple state abstractions). Finally, we demonstrate how this advantage can be leveraged in a guitar playing task, whereby an agent can reapply learned structure about the fret-board while learning a musical scale to quickly learn to play other scales that differ in transitions, rewards, and policy (Comparison to transferring successor features).

## Methods

An MDP [1] is a quintuple $M = \langle \mathcal{S}, \mathcal{A}, p, r, \gamma \rangle$ with a state space $\mathcal{S}$, an action space $\mathcal{A}$, a transition function $p : \mathcal{S} \times \mathcal{A} \times \mathcal{S} \to [0, 1]$, and a reward function $r : \mathcal{S} \times \mathcal{A} \times \mathcal{S} \to \mathbb{R}$. (In this article, we will always refer to the full transition function $p$ of an MDP.) The interaction of an agent in a particular task is modeled by a sequence of transitions between different states. Each transition from state $s$ to state $s'$ is initiated by an action $a \in \mathcal{A}$ and is rewarded with a scalar number $r(s, a, s')$. The probability of reaching state $s'$ from state $s$ using action $a$ is specified by the transition function $p(s, a, s')$. How strongly short-term rewards are favored over long-term rewards is controlled by the discount factor $\gamma \in [0, 1)$.

In *model-free learning*, for example Q-learning [19], the optimal decision-making strategy, called a *policy*, is learned through trial and error interactions in an MDP. Throughout these interactions, a policy is incrementally improved. During learning, only the policy $\pi$ and some form of cached values of the policy $\pi$ are stored at any point in time. In other words, the agent only learns and represents the net predicted reward value of an action in a given state, without needing to represent the specific outcomes of each action in terms of the subsequent states that will be encountered. In *model-based learning* [20, 21] an agent attempts to build a model of the task's transition and reward function and uses this model to predict sequences of future reward

outcomes $(r_1, r_2, \ldots)$ given a start state $s$ and a particular sequence of actions $(a_1, a_2, \ldots)$. While more computationally intensive, this approach is orthogonal to model-free learning, because using this model an RL agent can predict the value of any arbitrary policy, and it can flexibly adjust its policy if the reward changes. In this case, the agent's "knowledge" is sufficient to generalize across the space of all possible policies [20].

For example, in an Atari game [22], the number of states in an MDP is large and it may be inefficient to learn and memorize an optimal action for every possible image pixel configuration. In this case, state abstractions [6, 23] provide a framework for simplifying the input space into a lower-dimensional latent space. A state abstraction, also called state representation, is a function $\phi$ mapping the state space $\mathcal{S}$ to some other latent space. Because state representations are many-to-one relations, they can map different states to the same latent state and create a partitioning of the state space. A state partition is a subset of the state space that maps to the same latent or abstract state. An agent using a state abstraction $\phi$ operates on the space of state partitions and generalizes knowledge learned in one state across the entire state partition. For example, in Q-learning, a value update is applied to the entire state partition even if the update is computed only from one specific state transition, resulting in faster learning if the state abstraction is appropriate [4, 5].

While several approaches exist for constructing a useful state abstraction $\phi$ for complex MDPs (e.g., Atari games), this article investigates which state abstractions facilitate re-use across different tasks. Specifically, we consider the question of which algorithm should be used to learn a state abstraction $\phi$ from a hypothesis space of all possible state abstractions $\mathcal{H}$ to maximize the agent's ability to reuse knowledge in future tasks. A state abstraction is a function mapping states to a smaller latent abstract state space $\mathcal{S}_\phi$. The state-abstraction hypothesis space is then

$$\mathcal{H} = \{\phi : \mathcal{S} \to \mathcal{S}_\phi\} \tag{1}$$

and a representation learning algorithm searches this space to identify a state abstraction $\phi$. An agent that uses a state abstraction $\phi$ operates directly on the latent space $\mathcal{S}_\phi$ rather than the underlying state space $\mathcal{S}$. Depending on how $\phi$ constructs the latent state space $\mathcal{S}_\phi$, the agent may or may not be able to distinguish between a rewarding and a non-rewarding state.

Fig 1 presents an example of how state representations simplify a task. The column world task (Fig 1A) is a grid-world navigation task where an agent only receives reward by entering the right column. For this task, Fig 1B illustrates a reward-predictive state representation that generalizes across different columns, as indicated by the colouring. In this case, the $3 \times 3$ grid world is compressed into a smaller $3 \times 1$ grid world where a reward is given for entering the latent state $\phi_3$ (green) and no reward is given for the latent states $\phi_1$ (blue) and $\phi_2$ (orange). While the compressed version of the grid world does not preserve all information about the task, it still possible to predict future reward outcomes. For example, the path indicated by the black boxes is mapped to a latent state sequence of $(\phi_1, \phi_2, \phi_2, \phi_3)$. This latent state sequence could then be mapped to a reward sequence of $(0, 0, 0, 1)$. In this example, every possible path in the original task is mapped to a path in the compressed task that produces the same reward sequence. Hence, the smaller compressed task can be used to predict future reward outcomes of the original task and the state representation is thus reward predictive.

A reward-predictive state abstraction $\phi \in \mathcal{H}$ allows an agent to best predict which expected reward sequence $r_1, \ldots, r_t$ will be observed after executing a decision sequence $a_1, \ldots, a_t$ starting at a specific state $s$. If the random variable $R_t$ describes the reward that is observed after

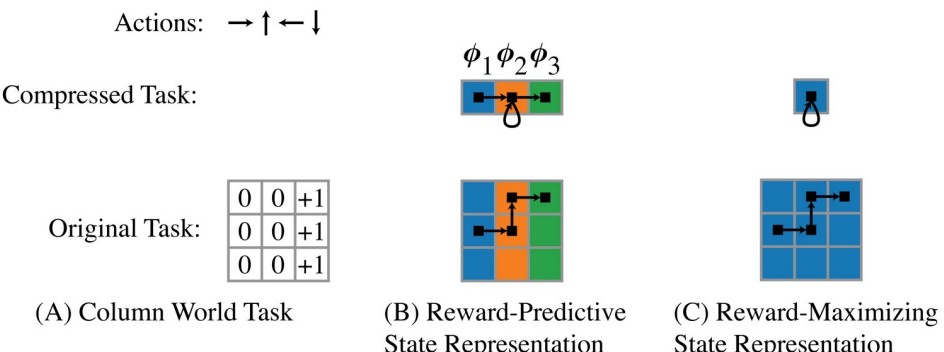

**Fig 1. State-abstraction examples, adopted from [9].** (A) The column world task is a 3 × 3 grid world where an agent can move up (↑), down (↓), left (←), or right (→). A reward of +1 is given when the right column is entered from the centre column by selecting the action "move right" (→). (B) A reward-predictive state representation generalizes across columns (but not rows) and compresses the 3 × 3 grid world into a 3 × 1 grid world with three latent states labelled with $\phi_1$, $\phi_2$, and $\phi_3$. In this compressed task, only the transition moving from the centre orange state $\phi_2$ to the right green state $\phi_3$ is rewarded. (C): A reward-maximizing state representation compresses all states into one latent state. In the 3 × 3 grid, there are three out of nine locations where an agent can receive a reward by selecting the action move right (→). If states are averaged uniformly to construct the one-state compressed task, then the move right action is rewarded with 1/3 and all other actions are not rewarded. In this case, an optimal policy can still be found using the compressed task, but accurate reward predictions are not possible.

following the action sequence $a_1, \ldots, a_t$ starting at state $s$, then the expected reward sequence is

$$(r_1, ..., r_t) = \mathbb{E}[(R_1, ..., R_t) \,|s, a_1, ..., a_t]. \tag{2}$$

The expectation in Eq (2) is conditioned on the start state $s$ and is computed over all possible trajectories in an MDP that follow the action sequence $a_1, \ldots, a_t$. A reward-predictive state abstraction needs to satisfy for any start state $s$ and action sequence $a_1, \ldots, a_n$ that

$$\mathbb{E}[(R_1, ..., R_t) \,|s, a_1, ..., a_t] = \mathbb{E}[(R_1, ..., R_t) \,|\phi(s), a_1, ..., a_t]. \tag{3}$$

The expectation on the right-hand side is conditioned on the latent state $\phi(s)$. Consequently the expectation on the right-hand side is computed over all possible trajectories in a latent space, while the expectation on the left-hand side is computed over all possible trajectories in an MDP [9]. While this model is different to learning an explicit approximation of the MDP's transition and reward function, learning a reward-predictive state abstraction is akin to model-based RL, because both systems are used to predict sequences of future reward outcomes.

Fig 1C presents a reward-maximizing state representation. Because the optimal action is to move right in this specific task, the task can be compressed into a single state in this example. If rewards are only given when an agent enters the right column by moving right, then only the right action is directly rewarded. In this case, the compressed task can still be used to find an optimal policy, because only the move right action is rewarded; the remaining three actions are not rewarded in any case. Nevertheless, the compressed task cannot be used to make accurate predictions of future reward outcomes, because this state representation simplifies the task into only one latent state (the agent does not know which column it is in: it is as if moving right simply produces stochastic rewards). Because this state abstraction allows an agent to recover the optimal policy, this state abstraction is reward maximizing in this example.

In this article, state abstractions are generated in one of two ways:

1. Enumerate all possible state abstractions using Algorithm U [24]. This method is used in Figs 3 and 6.

2. Learning a state-abstraction function from transition data. This method is used in Figs 7 and 8.

Reward-predictive state abstractions can be learned using Linear Successor Feature Models (LSFMs) [9] as a means to learn reward-predictive state abstractions. Successor Features (SFs) [25] are a generalization of the SR [10], and predict the expected visitation frequencies in some latent feature space:

$$\boldsymbol{\psi}^\pi(s, a) = \mathbb{E}[\phi(s_1) + \gamma\phi(s_2) + \gamma^2\phi(s_3) + \cdots | s = s_1, a = a_1, \pi], \tag{4}$$

where the expectation is computed over all infinite length trajectories that start in state $s$ with action $a$ and then follow the policy $\pi$. The discount factor $\gamma \in [0, 1)$ is used such that states in the more distant future are weighted to a lesser degree in the summation in Eq (4). Intuitively, SFs incorporate information about which latent state features are observed along a trajectory, including their relative temporal positions, because for each time step $t$ a different weight $\gamma^t$ is associated with the latent state feature vector $\phi(s_t)$ (Eq (4)) (allowing the state abstraction to distinguish between a reward sequence "+1, −1" vs. "−1, +1", for example). LSFMs extract this temporal property from SFs and construct a state abstraction $\phi$ that is predictive of the order with which particular latent state features are observed. Critically, the LSFM latent space is constructed so as to most efficiently predict reward sequences without being tied to the specific transitions or rewards, and thus permit a "deeper" form of transfer loosely akin to analogical reasoning.

If such a state abstraction also associates each feature vector $\phi(s_t)$ with one-step reward outcomes, then this state abstraction is reward-predictive. Mathematically, this intuition can be generalized to predict reward outcomes for any start state $s$ and action sequence $a_1, \ldots, a_t$ and indeed, learning LSFMs is equivalent to learning reward-predictive state abstractions [9]. Please refer to supporting S3 Text for a detailed description of how LSFMs are used.

Note that this LSFM approach contrasts with the typical application of SFs in which Q-values are expressed as a dot-product between the SF vectors $\boldsymbol{\psi}^\pi(s, a)$ and a reward-model vector. While that approach allows an agent to re-use SFs when rewards and the associated reward-model vector change, it does not afford analogical transfer when transitions change. In fact, because SFs depend on the transition function and a particular policy, a transferred SF has to be relearned and adjusted to a specific task. In contrast to SFs, reward-predictive state abstractions are independent of a specific policy and can be used to generalize across all policies that are defined in terms of the latent states. More concretely, a reward-predictive state abstraction can be used to predict the value of any arbitrary abstract policy by first predicting which reward sequence a specific policy generates and then computing the discounted sum over this reward sequence [9, Theorem 4]. Fig 2 presents an intuitive transfer example and plots different SFs for each task. Consequently, an agent would have to adjust a previously learned SF.

An alternative to using LSFMs are Linear Action Models (LAM), which predict the expected next state instead of SFs. LAMs are very closely tied to LSFMs and can also be used to learn reward-predictive state abstractions [9]. Because we found that LSFMs are easier to use than LAMs in practice, this article focuses on LSFMs. Please refer to S2 Text for a more detailed description of LSFMs and the connection to LAMs.

## Results

To generalize knowledge across different tasks, a compressed state abstraction is needed that preserves key aspects of the tasks even if the details of transitions or rewards change. Consider the transfer example in Fig 2, where an agent is first presented Task A, and then transfers a state abstraction to Task B. The key similarity between the tasks is evident in that they both have columnar structure, but transitions, rewards, and the optimal policy can all differ. In this example, the reward-predictive state abstraction (Fig 2B) can be re-used to plan a different policy in Task B, while the reward-maximizing state abstraction (Fig 2C) cannot be re-used in Task B. Of course, such a benefit is only possible if the two tasks share an abstract relation: This columnar state abstraction would not be useful in subsequent MDPs that arranged in rows. Below, we consider how multiple state abstractions can be learned and where generalization involves an inference process to select which one of them is most applicable [3].

In principle, there always exists one state abstraction that is both reward-maximizing and reward-predictive in a model-based agent: Trivially, if the identity map is used to map nine distinct states into a latent space of nine distinct states, then such a state representation is always reward maximizing and reward predictive. However, such a state representation is not really "abstract" in that it does not inform an agent across which states information can be generalized. But, for the same reason, this representation preserves information that might be needed in other tasks. We will further discuss this trade-off in the context of our online learning simulations in the following section (Learning to transfer multiple state abstractions).

### Transfer with single state abstractions

The above example was illustrative for a single MDP designed to show the potential utility of reward-predictive state abstractions. We next systematically assess the generalization potential of reward-maximizing or reward-predictive state abstractions across a range of different tasks. The goal of this experiment is to be algorithm agnostic: Rather than focusing on how a particular algorithm performs at transfer with a single learned state abstraction, we enumerated the entire hypothesis space $\mathcal{H}$ for all possible partitions of the state space and evaluated them in all transfer tasks. (Out of a set of tasks, one task was randomly chosen for evaluation of a single state abstraction $\phi$. Subsequently, this state abstraction $\phi$ is evaluated in all other remaining transfer tasks. In all simulations, the evaluation and transfer tasks are distinct.) For each state abstraction in $\mathcal{H}$, we computed a compressed abstract MDP [23] for every tested MDP and solved it using value iteration [1, Chapter 4.4] (please also refer to supporting S1 Text). A reward-maximizing state abstraction is then identified by testing the computed policy in a single randomly selected task for $N$ trials over $T$ time steps and computing the total reward

$$R_{\text{total}} = \frac{1}{N} \sum_{n=1}^{N} \sum_{t=1}^{T} r_{n,t}, \tag{5}$$

where $r_{n,t}$ is the reward incurred in trial $n$ at time step $t$. A reward-predictive state abstraction is identified by sampling $N$ random state-and-action sequence pairs $(s_n, a_{n,1}, \ldots, a_{n,T})$ and predicting the reward sequence $\hat{r}_{n,1}, \ldots, \hat{r}_{n,T}$ using the abstract MDP. The reward-sequence prediction error is

$$RS_{\text{error}} = \frac{1}{N} \sum_{n=1}^{N} \sum_{t=1}^{T} |r_{n,t} - \hat{r}_{n,t}|. \tag{6}$$

We considered three types of tasks: column-worlds (like those in the motivating example), 100 randomly generated MDPs, and grid worlds (Fig 3A, 3B and 3C). For each transfer

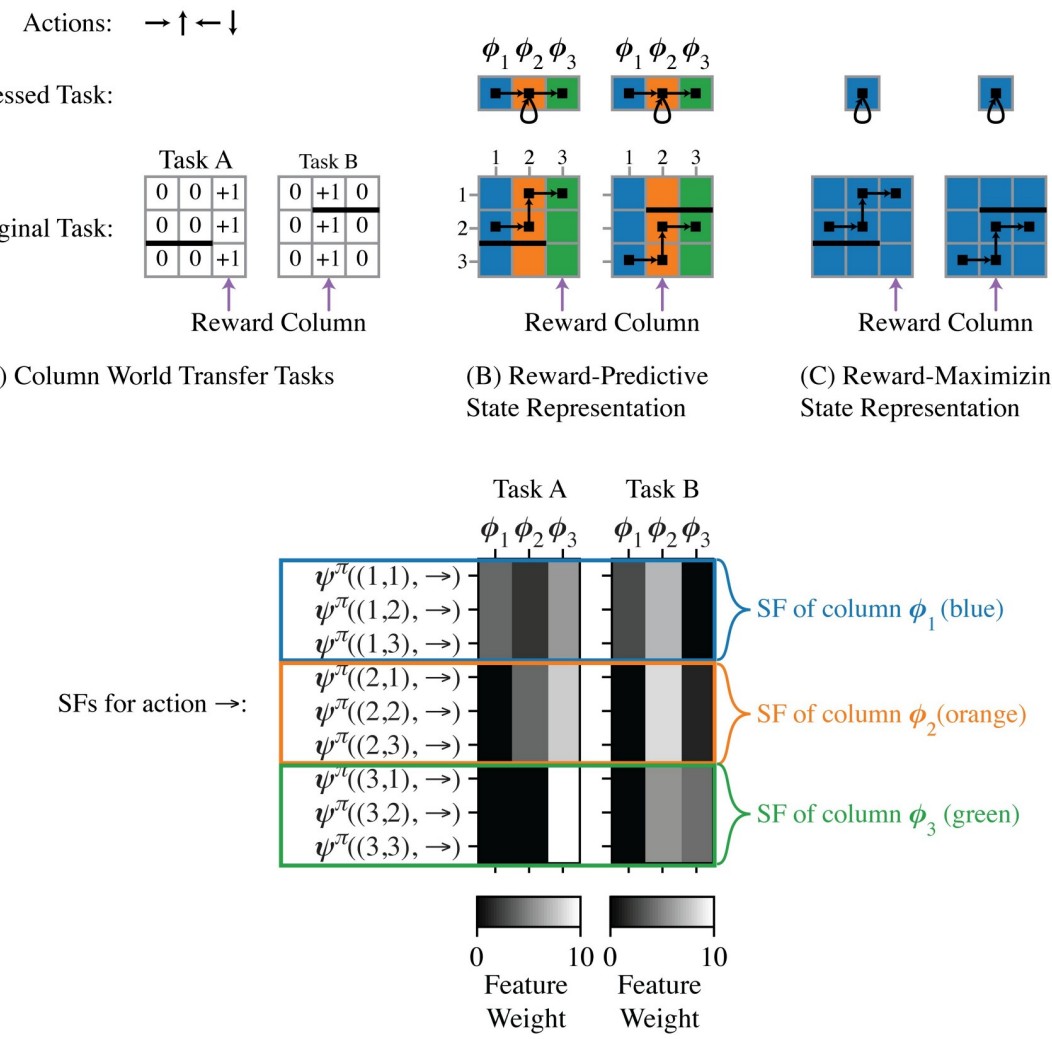

(D) Reward-predictive state representations merge states with equal successore features into the same latent state.

**Fig 2. Transferring state abstractions between MDPs.** (A) In both grid-world tasks, the agent can move up ($\uparrow$), down ($\downarrow$), left ($\leftarrow$), or right ($\rightarrow$) and is rewarded when a reward column is entered. The black line indicates a barrier the agent cannot pass. Both Task A and Task B differ in their rewards and transitions, because a different column is rewarded and the barrier is placed at different locations. (B) A reward-predictive state representation generalizes across different columns and the corresponding SFs are plotted below in (D). (D) Each row in the shown matrix plots visualizes the entries of a three dimensional SF vector. Similar to the example in Fig 1, a reward-predictive state abstraction merges each column into one latent state, as indicated by the colouring. In both tasks, reward sequences can be predicted using the compressed representation for any arbitrary start state and action sequence, similar to Fig 1B. In this case the agent simply needs to learn a different policy for Task B using the same compressed representation. In contrast, the matrix plots in the bottom panels illustrate that SFs are different in each task and cannot be immediately reused in this example (because SFs are computed for the optimal policy which is different in each task [14]). Note that states that belong to the same column have equal SF weights (as indicated by the coloured boxes). LSFMs construct a reward-predictive state representation by merging states with equal SFs into the same state partition. This algorithm is described in supporting S3 Text and prior work [9]. (C) One possible reward-maximizing state abstraction may generalize across all states. While it is possible to learn or compute the optimal policy using this state abstraction in Task A (i.e., always go right), this state abstraction cannot be used to learn the optimal policy in Task B in which the column position is needed to know whether to go left or right. This example illustrates that reward-predictive state representations are suitable for re-use across tasks that vary in rewards and transitions. While reward-maximizing state abstractions may compress a task further than reward-predictive state abstractions, reward-maximizing state abstractions may also simplify a task to an extend that renders them proprietary to a single specific task.

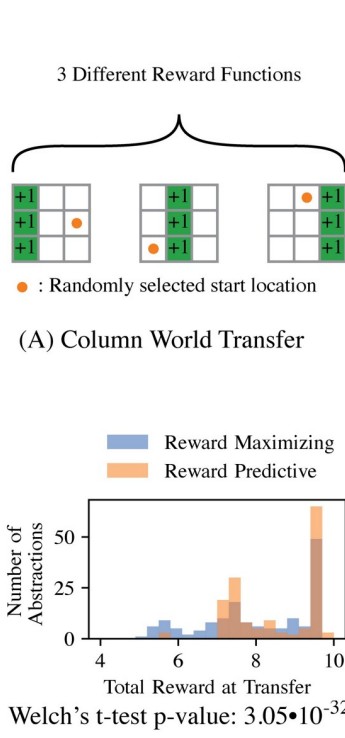

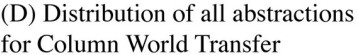

(A) Column World Transfer

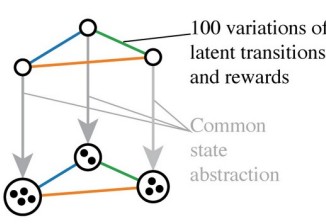

(B) Random MDP Transfer

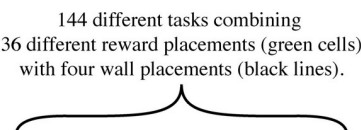

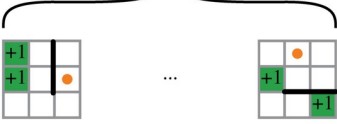

(C) Grid-World Navigation Transfer

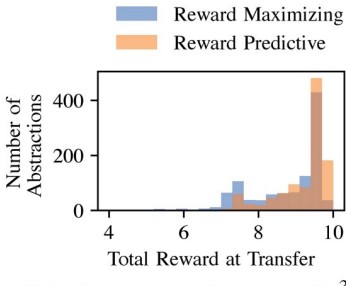

Welch's t-test p-value: $3.05 \cdot 10^{-32}$

(D) Distribution of all abstractions
for Column World Transfer

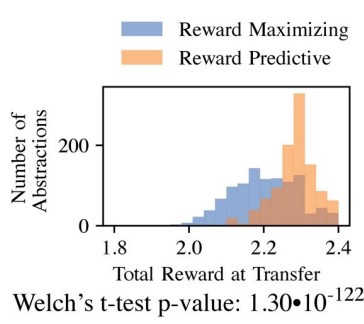

Welch's t-test p-value: $1.30 \cdot 10^{-122}$

(E) Distribution of all abstractions
for Random MDP Transfer

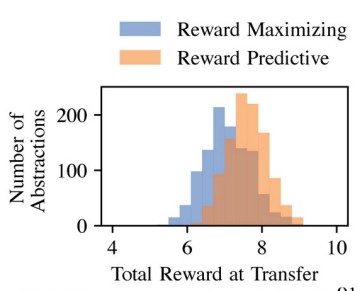

Welch's t-test p-value: $3.00 \cdot 10^{-91}$

(F) Distribution of all abstractions
for Grid-World Navigation Transfer

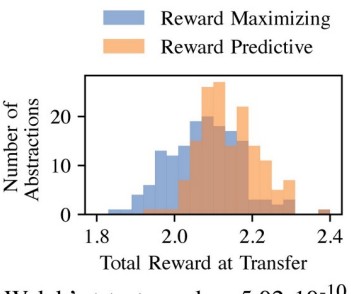

Welch's t-test p-value: $1.41 \cdot 10^{-3}$

(G) Distribution for abstractions
with three latent states for
Column World Transfer

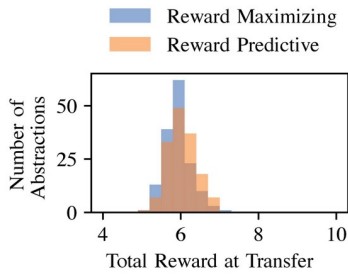

Welch's t-test p-value: $5.02 \cdot 10^{-10}$

(H) Distribution for abstractions
with three latent states for
Random MDP Transfer

Welch's t-test p-value: $6.19 \cdot 10^{-3}$

(I) Distribution for abstractions
with three latent states for
Grid-World Navigation Trasnfer

**Fig 3. Minimizing reward-sequence prediction errors identifies state abstractions amenable for "deep transfer".** For each task set (A, B, C), all possible state abstractions in $\mathcal{H}$ were enumerated using Algorithm U [24] to obtain a ground truth distribution over the hypothesis space $\mathcal{H}$. In each grid-world task (A, C) the agent can transition up, down, left, or right to move to an adjacent grid cell. If the agent attempts to transition of the grid or across one of the black barriers in (C), then the agent remains at its current grid position. State abstractions were scored by compressing an MDP using the state abstraction of interest [6]. The total reward score was computed by running the computed policy 20 times for 10 time steps in the MDP from a randomly selected start state. The reward-sequence error was computed by selecting 20 random start states and then performing a random walk for 10 time steps. (D, E, F) The histograms report averages over all repeats and transfer MDPs for all state abstractions that are possible in a nine state MDP. (G, H, I) The histograms report averages over all repeats and transfer MDPs for all state abstractions that compress nine states into three latent states. For each histogram, the Welch's t-test was performed to compute the p-values of the difference in mean total reward being insignificant for each histogram.

experiment, all possible state abstractions are enumerated and the top 5% scoring state abstractions were re-evaluated on the remaining transfer MDPs and the total rewards generated by these state abstractions are plotted as histograms in Fig 3. In all cases, state abstractions with low reward-sequence prediction errors $RS_{\text{error}}$ generate a higher total reward at transfer than state abstractions that were selected based on their ability to construct a well performing policy (produce a high $R_{\text{total}}$ score) on the original MDP. Note that restricting the hypothesis space $\mathcal{H}$ to abstractions that construct three latent states (second row of histograms in Fig 3) does not change the overall result. This result indicates that reward-predictive state abstractions encode information about an MDP that can be generalized across different MDPs that share the same abstract structure. In the following two sections (Transfer with multiple state abstractions and Learning to transfer multiple state abstractions), we will present an extension to environments in which multiple structures are possible and have to be inferred.

Fig 3D and 3G present the results for the transfer experiment discussed in Fig 2. Both histograms indicate that state abstractions with low reward-sequence prediction errors outperform on average state representations that only maximize total reward in one of the tasks. Because all three MDPs can be compressed into three latent states, constraining the hypothesis space $\mathcal{H}$ to only contain state abstractions that create three latent states does not impact the total reward generated at transfer time significantly. In this case, both histograms have equal support.

To further control for a potential dependency between the constructed MDPs and a particular state-representation type, the experiment in Fig 3B randomly generates transition and rewards. This experiment is similar to the previous test case in that all 100 randomly generated MDPs can be compressed with the same state representation. These MDPs are constructed by generating random three-state MDPs and then "inflating" the state space size based on this common but randomly generated state abstraction. Aside from this common "hidden" state representation, these 100 MDPs differ in both transition and reward functions. The histograms confirm the claim that abstractions yielding low reward-sequence prediction errors perform best in generalization across different MDPs (Fig 3E and 3H). In contrast, state representations that result in high total reward in any of the original MDPs generate on average less reward in any of the remaining MDPs. Again, constraining the hypothesis space $\mathcal{H}$ to only include abstractions that construct three latent states does not change the support of the histogram in Fig 3H, but the shape changes and the median shifts. This shift can be explained by the fact that incorrectly compressing a task and incurring approximation errors can quickly degrade an agent's ability to perform optimally. If a state abstraction does not maximally compress a task, for example from nine to eight states, then performance may not degrade as quickly.

The above simulation assumed lossless compression (given that the abstraction was selected and then inflated to generate larger state spaces). To test which state abstractions generalize across tasks when no "hidden" state abstraction is embedded in the tasks, we next considered situations in which state spaces could not be compressed without some information loss. Fig 3C presents a transfer experiment where two reward locations and four different wall placements are permuted in a grid world. These changes in reward and wall locations resemble changes in the transition and reward functions. In this experiment, the MDPs cannot be compressed without incurring some loss, because the grid location is important for predicting where the goal locations are and what action is optimal at each location. However, both histograms in Fig 3F and 3I indicate that state abstractions that minimize the reward-sequence prediction error criterion still perform better than those that maximize total reward. By nature, grid worlds have a specific topology of the state space and state representations that cluster only neighbouring states approximately preserve the grid location information and would be expected to perform relatively well across all MDPs. If the hypothesis space $\mathcal{H}$ is constrained to abstractions that compress nine states into three latent states, then the advantage shrinks. This

difference can be explained by the fact that for arbitrary navigation tasks grid worlds should ideally not be compressed (for efficient navigation an agent needs to be aware of its position), and hence neither abstraction yields an optimal policy. However, the histogram in Fig 3F suggests that there exist several state abstractions that compress nine states into six or seven latent states that can still lead to (close to) optimal performance.

Note that the identity map, which does not compress the state space and can always be used to construct an optimal policy, is also included in this histogram and occurs in the bin with highest total reward. The identity map is included exactly once into each histogram that plots the distribution for all abstractions, because this experiment tests each possible state abstraction once. This experiment highlights a trade-off between the ability to obtain an optimal policy in a task and re-use of a particular state abstraction that compresses a task.

## Transfer with multiple state abstractions

The previous experiment assumes that all tasks share a common "hidden" state abstraction that can be learned and re-used by an agent. In this section, we consider the situation in which different MDPs might correspond to different abstractions. A non-parametric Bayesian model maintains a belief space of possible state abstractions [3, 4], which it can use for inference. Fig 4 illustrates how the curriculum of tasks is randomly generated. This task curriculum is observed in sequence by the non-parametric Bayesian model and the model is signalled when a switch between tasks occurs. Each task can be compressed in one of two different ways (this approach can be expanded to larger numbers without loss of generality; two is used here for clarity of exposition). Critically, this state abstraction is hidden from the learning agent. After observing an MDP sequence $M_1, \ldots, M_t$, the agent updates its belief space $\mathcal{B}_t$ using a posterior over which state abstraction is most suitable to solve a given task $M_t$:

$$\Pr(\phi|M_t, \mathcal{B}_t, c_t) \propto \Pr(\phi|M_t)\Pr(\phi|\mathcal{B}_t, c_t), \qquad (7)$$

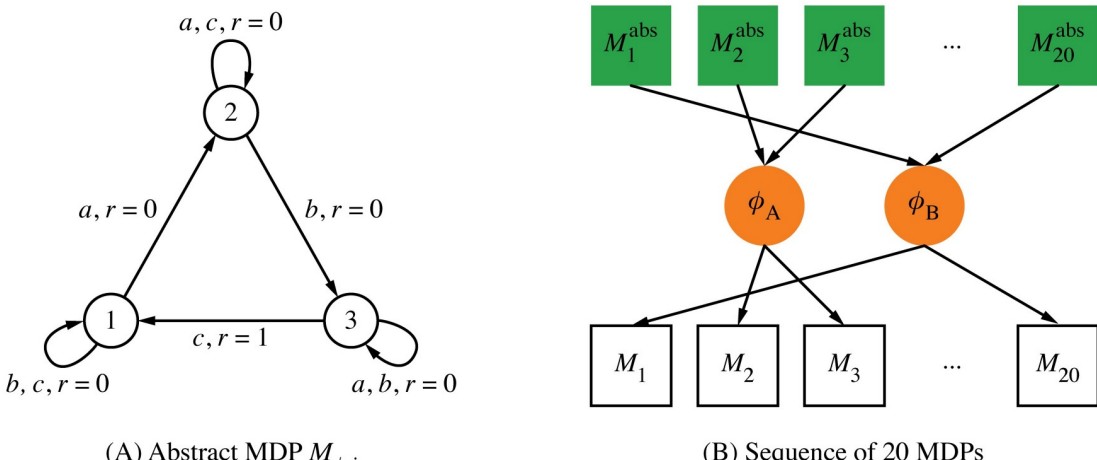

(A) Abstract MDP $M_{\phi,i}$          (B) Sequence of 20 MDPs

**Fig 4. Transfer with multiple state abstractions curriculum.** (A) A curriculum of transfer tasks is generated by first constructing the three-state MDP. At each state, only one action causes a transition to a different state. Only one state-to-state transition is rewarded; the optimal policy is to select the correct action needed to cycle between the node states. (B) To generate a sequence of abstract MDPs $M_1^{abs}, \ldots, M_{20}^{abs}$, the action labels are randomly permuted as well as the transitions generating positive reward (similar to the Diabolical Rooms Problem [3]). Two hidden state abstractions $\phi_A$ and $\phi_B$ were randomly selected to "inflate" each abstract MDP to a nine-state problem. One state abstraction was used with a frequency of 75% and the other with a frequency of 25%. The resulting MDP sequence $M_1, \ldots, M_{20}$ was presented to the agent, without any information about which state abstraction was used to construct the task sequence.

where $c_t(\phi)$ is the count of how often an abstraction $\phi$ was used in the previous $t - 1$ tasks. These counts are used to construct a Chinese Restaurant Process (CRP) [26, 27] prior for an intensity $\alpha > 0$:

$$\Pr(\phi|\mathcal{B}_t, c_t) = \begin{cases} \dfrac{c_t(\phi)}{t - 1 + \alpha} & \text{if } \phi \in \mathcal{B}_t \\[2ex] \dfrac{\alpha}{t - 1 + \alpha} & \text{otherwise.} \end{cases} \tag{8}$$

The posterior is also conditioned on the MDP $M_t$ through the factor $\Pr(\phi|M_t)$. Using a loss function $l$ (we consider both reward-maximizing and reward-predictive losses), each state abstraction $\phi$ can be scored and for $\beta \geq 0$, the probability of this state abstraction being suitable to solve $M_t$ is the soft-max probability

$$\Pr(\phi|M_t) \propto e^{-\beta l(\phi)}. \tag{9}$$

To determine which state abstraction should be added into the abstraction belief set $\mathcal{B}_t$, the non-parametric Bayesian agent has access to the best scoring state abstraction $\phi_{\text{next-best}}$ not included into $\mathcal{B}_t$. The posterior $\Pr(\phi|M_t, \mathcal{B}_t, c_t)$ is computed over the set of state abstractions $\mathcal{B}_t \cup \{\phi_{\text{next-best}}\}$. (The goal is not to design an algorithm that can solve a sequence of tasks efficiently, but to analyze which state abstractions generalize across different tasks. Thus, for the moment, we assume that the agent has access to an oracle that knows the transition function of each new MDP and can score the loss for each compression. Using an oracle that tabulates all possible state abstractions $\phi \in \mathcal{H}$ gives insight into which state abstractions generalize across different tasks, while being algorithm agnostic; below we relax the need for an oracle.) In contrast to the previously presented simulation, this non-parametric Bayesian agent is constrained to only use state abstractions that compress nine-state MDPs to three-state MDPs. Consequently, the model is forced to generalize across different states and cannot default to only using the identity state abstraction, which does not compress an MDP and is both reward-predictive and reward-maximizing. If $\alpha$ increases, the resulting prior and posterior assign a higher probability to adding the next-best state abstraction $\phi_{\text{next-best}}$ into $\mathcal{B}_t$. In this case, the CRP prior influences the posterior more strongly. If $\beta$ increases, then more emphasis is given on using the loss function $l$ to determine which state abstraction should be used from the set $\mathcal{B}_t \cup \{\phi_{\text{next-best}}\}$ and the CRP prior is effectively ignored.

Rather than using the empirical scores $R_{\text{total}}$ or $RS_{\text{error}}$, the agent is allowed to observe a tabulation of all possible transitions and rewards to obtain a ground truth score for each abstraction. In these experiments, reward-maximizing state abstractions are identified by assessing how much using a state abstraction impacts the value of the policy $\pi_\phi$ relative to that of the optimal policy in the abstract MDP:

$$l_{\text{maximizing}}(\phi) = \max_{s \in \mathcal{S}} \left( V^{\pi^*}(s) - V^{\pi_\phi}(s) \right), \tag{10}$$

where $V^{\pi^*}$ is the optimal discounted value function [1], $\pi^*$ is the optimal policy, and $V^{\pi_\phi}$ is the discounted value function of the policy $\pi_\phi$ evaluated in the task itself. Reward-predictive state abstractions are scored by the loss function $l_{\text{predictive}}$ bounding the reward-sequence prediction error

$$\forall i, \forall a_1, ..., a_t, \ l_{\text{predictive}}(\phi)C_{\gamma,t} \geq |\mathbb{E}[r_t|i, a_1, ..., a_t] - \hat{r}_t|, \tag{11}$$

where $C_{\gamma,t}$ is a constant that depends on the action-sequence length $t$ and discount factor $\gamma$. The loss function $l_{\text{predictive}}$ is computed using the SF model [9]. Supporting S2 Text presents all

details on how to compute $l_{\text{predictive}}$. If any of the two loss functions evaluates to zero for a state abstraction $\phi$, then $\phi$ is either a globally optimal reward-maximizing or reward-predictive state abstraction. For reward-predictive state abstractions, this property holds because each tested task in Fig 1 can be compressed (by construction) and the LSFM Bisimulation Theorem [9, Theorem 2] applies. In this case, if $l_{\text{predictive}} = 0$, then the state abstraction $\phi$ can be used to predict reward-sequences accurately. (Alternatively, one could also use $RS_{\text{error}}$ as defined in Eq (6).).

Fig 5 plots the results from testing the agent with each loss function for various $\alpha$ and $\beta$ settings. The agent selects its policy by using the posterior to mix the policies that would be optimal in the respective abstract MDPs (as described in the previous section, policies are computed using value iteration on the abstract MDP). Setting $\beta = \infty$ means that the probability $\Pr(\phi|M_t)$ is deterministic: The highest scoring state abstraction is assigned a probability of

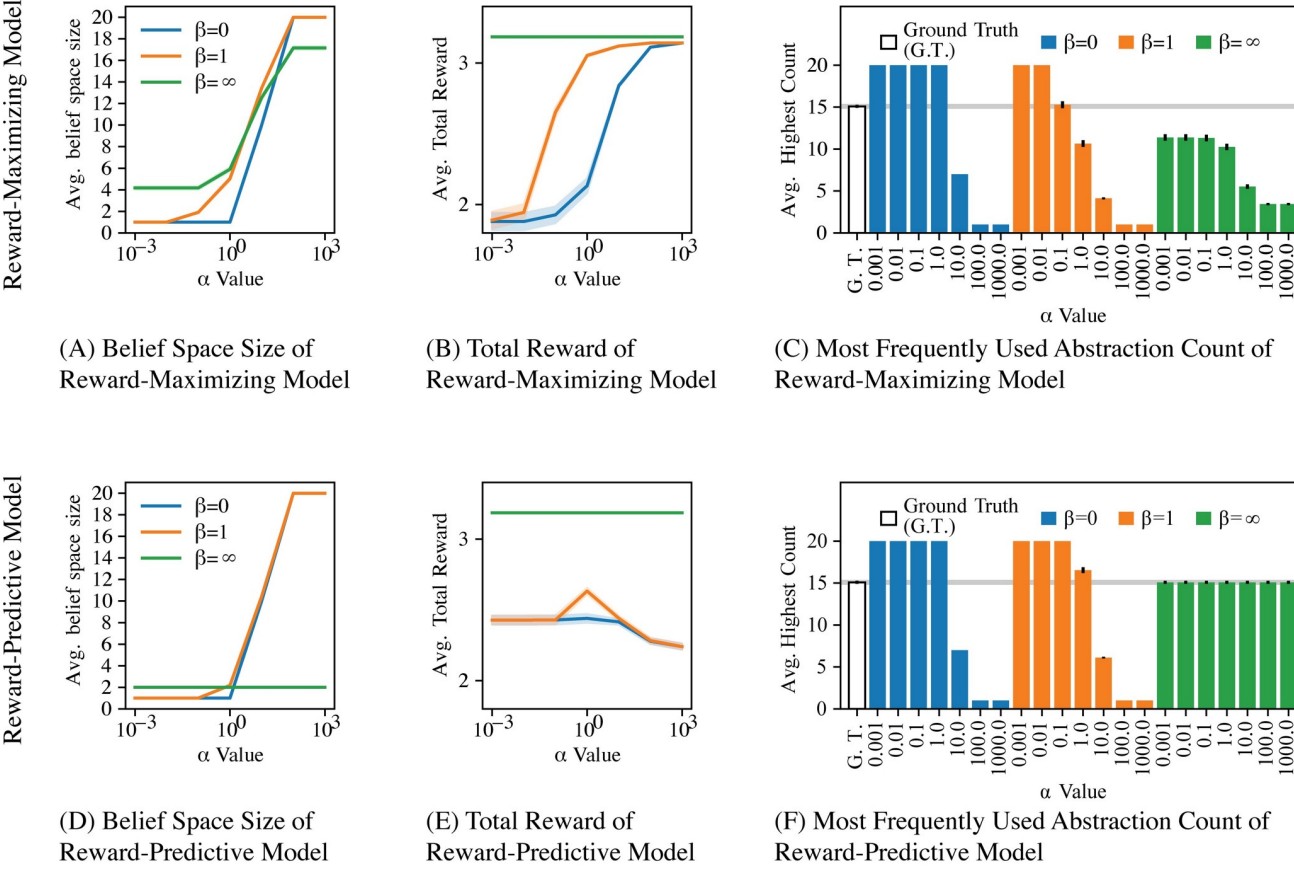

(A) Belief Space Size of Reward-Maximizing Model

(B) Total Reward of Reward-Maximizing Model

(C) Most Frequently Used Abstraction Count of Reward-Maximizing Model

(D) Belief Space Size of Reward-Predictive Model

(E) Total Reward of Reward-Predictive Model

(F) Most Frequently Used Abstraction Count of Reward-Predictive Model

**Fig 5. Results for transfer with multiple state abstractions experiment.** (A, D) Plot of how different $\alpha$ and $\beta$ model parameters influence the average size of $\mathcal{B}_t$ after training. (B, E) Performance of each model (average total reward per MDP) for different $\alpha$ and $\beta$ model parameters. After observing the transition and reward tables of a task $M_t$ in the task sequence, the average total reward was obtained by first computing a compressed abstract MDP for each abstraction and then solving each compressed MDP using value iteration, as described in supporting S1 Text. The resulting mixture policy was then tested in the task $M_t$ for 10 time steps while logging the sum of all obtained reward. If $\beta = \infty$ the agent obtains an optimal total reward level when using either loss function for ten time steps in each MDP. (C, F) Plot of the average count for the most frequently used state abstraction. As described in Fig 4, one of two possible "hidden" state abstractions, $\phi_A$ and $\phi_B$, were embedded into each MDP. Each task sequence consists of 20 MDPs and on average 15 out of these 20 MDPs had the state abstraction $\phi_A$ embedded and the remaining MDPs had the state abstraction $\phi_B$ embedded. The white bar labelled "Ground Truth" plots the ground-truth frequency of the "hidden" state abstraction $\phi_A$. If the non-parametric Bayesian model correctly detects which state abstraction to use in which task, then the average highest count will not be significantly different from the white ground truth bar. In total, 100 different task sequences, each consisting of 20 MDPs, were tested and all plots show averages across these 100 repeats (the standard error of measure is indicated by the shaded area and variations are very low if not visible).

one and all other state abstractions are assigned a probability of zero. This case is equivalent to only using the loss function to select a state abstraction while ignoring the CRP prior $\Pr(\phi|\mathcal{B}_t, c_t)$, because the factor $\Pr(\phi|M_t)$ is either zero or one in Eq (7). For low $\beta$ settings, the prior is used to determine which state abstraction is used. If $\alpha$ is high, then up to 20 state abstractions are added into the belief set $\mathcal{B}_t$. Because the prior influences the posterior heavily, the total reward of the resulting agent is comparably low, because the agent is not well informed about which state abstraction should be used on a given task. For $\beta = \infty$, the loss function influences the posterior strongly.

The key difference between the two loss functions becomes apparent when analyzing how the agent maintains the belief space $\mathcal{B}_t$. Using the loss function $l_{\text{predictive}}$, which identifies reward-predictive state abstractions, the agent identifies the correct ground truth state abstractions that were used to generate the task sequences. Fig 5F shows that the agent correctly learns that one state abstraction occurs with a frequency of 75%. Because the agent only maintains two belief abstractions, the agent correctly estimates that the other abstraction occurs with a frequency of 25%.

In contrast, when the loss function $l_{\text{maximizing}}$ is used, Fig 5A and 5B demonstrate that the agent can only achieve optimal reward by isolating a significantly higher number of state abstractions than the reward-predictive model. At best, using $l_{\text{maximizing}}$ and a small $\alpha$ value the agent is capable of isolating between four and five state abstractions. For high $\alpha$ settings, the agent effectively memorizes a solution for almost every task by increasing the size of its belief set $\mathcal{B}_t$, because a previously used state abstraction does not generalize to the next task. The model is thus able to achieve optimal reward only if it constructs a new reward-maximizing abstraction for each MDP. Note that this experiment does not account for any cost associated with learning or constructing a state abstraction for each task from scratch. In the following section, this assumption is removed and the presented results illustrate how constructing a reward-maximizing state abstraction results in slower learning. When using the loss function $l_{\text{predictive}}$, the agent can correctly identify which state abstraction to use for which MDP and obtain an optimal reward level while only using two different state abstractions (green curves in Fig 5D and 5E). This confirms the claim that reward-predictive state abstractions generalize across different tasks.

## Learning to transfer multiple state abstractions

While the previous transfer experiment presents evidence that reward-predictive state abstractions generalize across different tasks, for exposition these previously presented simulations assumed that a full tabulation of all transitions and reward is accessible for the agent to score the loss. In addition, it was possible to configure both reward-predictive and reward-maximizing models such that an optimal reward level is always obtained given the agent can always construct a new reward-maximizing abstraction. This section presents an experiment where an intelligent system has to learn through trial-and-error interactions with a novel sequential decision-making task thereby simultaneously learning the transitions and making inferences about which abstraction is appropriate to reuse, in a Bayesian mixture of experts scheme [28] that is updated after interacting for a certain number of episodes in a particular task. If an intelligent agent is capable of extracting a particular structure from one task to accelerate learning in another task, then this agent will generate more reward in certain tasks than an agent that does not transfer any latent structure.

In the following simulation experiments, an agent is allowed to interact with a task for a certain number of trials, called *episodes*. The interaction data itself is a data set of transition quadruples of the form $(s, a, r, s')$ that describe a transition from state $s$ to state $s'$ that occurs by selecting action $a$ and is rewarded with a scalar reward $r$.

The generalized non-parametric Bayesian model maintains a belief space of state abstractions $\mathfrak{B}_t$ that is updated after interacting for 200 episodes in a task. (We found that 200 episodes allow each tested algorithm to converge to an optimal policy.) Subsequently, the collected data is used to learn a new state abstraction. A reward-predictive state abstraction is obtained using the LSFM discussed in the Methods section. A reward-maximizing state abstraction is obtained by clustering states with approximately equal Q-values into latent states. Please refer to supporting S3 Text. for a detailed description of the algorithmic implementation and how hyper-parameters were tuned. After interacting with a task and learning a new state abstraction, the belief set $\mathfrak{B}_t$ is updated using the posterior probabilities $\Pr(\phi|M_t, \mathfrak{B}_t, c_t)$. During learning in the next task, the state abstractions stored in the belief set $\mathfrak{B}_t$ are used to generalize Q-values across different states during learning in a task. While the agent observes transition data in a task $M_t$, a separate Q-learning agent is maintained for each state abstraction $\phi \in \mathfrak{B}_t$ and another for the identity state abstraction $\phi_{\text{identity}}: s \mapsto s$. (The motivation here is that the agent should consider not only the Q-values of actions that pertain to previously seen abstractions but that it should also have potential to learn Q-values in the full observable state space. We consider biological implications of this assumption in the discussion). Using a state abstraction $\phi$, a state $s \in \mathcal{S}$ is mapped to a latent state $\phi(s)$ and this latent state $\phi(s)$ is given as input to the Q-learning algorithm. (If Q-learning would normally observe a transition $(s, a, r, s')$, the algorithm now observes a transition $(\phi(s), a, r, \phi(s'))$. Because Q-learning caches Q-values for latent states and multiple states map to the same latent state, the agent now generalizes Q-values across multiple states and can thus converge faster.) The Q-learning algorithm thus generalizes Q-values to multiple states that map to the same latent state. As in the Bayesian mixture of experts scheme [3, 28], the agent selects its overall policy by mixing the policies of each Q-learning agent using the posterior probabilities $\Pr(\phi|M_t, \mathfrak{B}_t, c_t)$. Specifically, the probability of selecting action $a$ at state $s$ is

$$\pi(s, a) = \sum_{\phi \in \mathfrak{B}_t \cup \{\phi_{\text{identity}}\}} \Pr(\phi|M_t, \mathfrak{B}_t, c_t)\pi_\phi(s, a), \tag{12}$$

where $\pi_\phi$ are the action-selection probabilities of the Q-learning algorithm corresponding to the state abstraction $\phi$. For example, if the posterior probabilities place a high weight on a previously learned state abstraction $\phi \in \mathfrak{B}_t$, then the agent will effectively select actions similar to a Q-learning algorithm that is run on the latent state space constructed by the state abstraction $\phi$. In this case, an optimal policy should be obtained more quickly in comparison to not using any state abstraction, assuming the state abstraction $\phi$ is constructed properly for the given task. If the posterior places a high weight on the identity state abstraction $\phi_{\text{identity}}$, then the model will effectively select actions similarly to the usual Q-learning algorithm.

While training, the non-parametric Bayesian model also uses all observed transitions $(s, a, r, s')$ to construct a transition and reward table. After 200 training episodes on a particular task, these transition and reward tables are used to construct either a reward-maximizing or reward-predictive state abstraction $\phi_{\text{next-best}}$. Then, the posterior probabilities $\Pr(\phi|M_t, \mathfrak{B}_t, c_t)$ are computed as described in Eq 7 and a state abstraction is sampled using this posterior distribution. Depending on the parameter settings for $\alpha$ and $\beta$, the newly learned state abstraction $\phi_{\text{next-best}}$ may be added into the belief set $\mathfrak{B}_{t+1}$ for the next task or previously learned state abstraction $\phi \in \mathfrak{B}_t$ is re-used and its count $c_t(\phi)$ is increased. When training on the first task, the belief set $\mathfrak{B}_1$ is initialized to the empty set.

This model is tested on the task sequence illustrated in Fig 6. The top row depicts two different maze maps that are used to construct a curriculum of five tasks. Each map is a $10 \times 10$

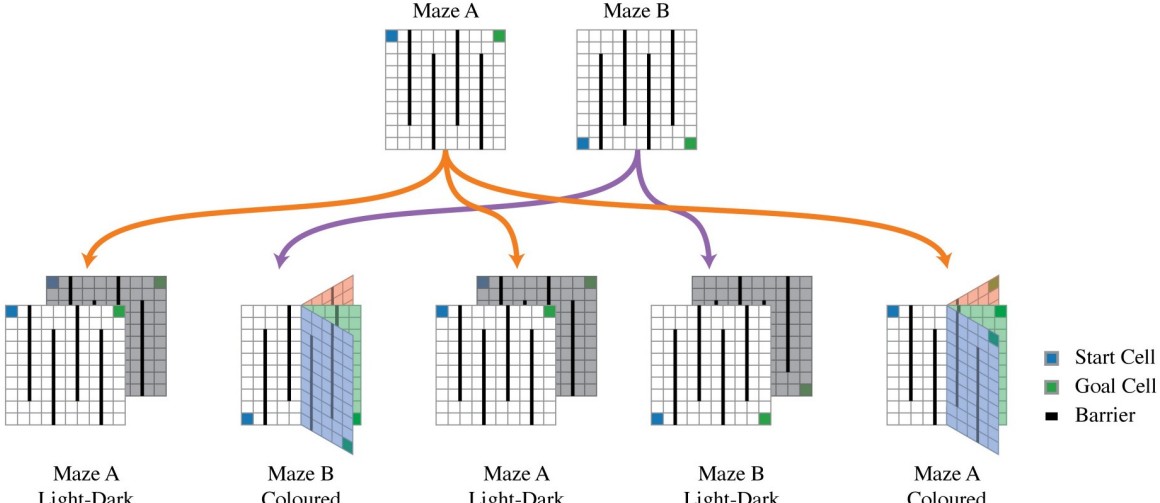

**Fig 6. Maze curriculum.** Maze A and Maze B are augmented with an irrelevant state variable to construct a five-task curriculum. In each maze, the agent starts at the blue grid cell and can move up, down, left, or right to collect a reward at the green goal cell. The black lines indicate barriers the agent cannot pass. Once the green goal cell is reached, the episode finishes and another episode is started. (These rewarding goal cells are absorbing states.) Transitions are probabilistic and succeed in the desired direction with probability 0.95; otherwise the agent remains at its current grid cell and cannot transition off the grid map or through a barrier. A five-task curriculum is constructed by augmenting the state space either with a "light" or "dark" colour bit (first, third, and fourth task), or the right half of the maze is augmented with the colour red, green, or blue (second and fifth task).

grid world where the agent has to navigate from the blue start state to the green goal location. Once the green goal location is entered, the agent receives a reward of +1 and the episode is ended. The transition dynamics are the same in each task with the difference that the agent cannot cross the black barrier. These two mazes are mirror images of another and the optimal action is different at each grid cell. Consequently, transitions, rewards, and the optimal policy of Maze A and Maze B are different at every state and cannot be immediately transferred from one maze to another.

Using these two mazes, a task sequence is constructed by adding a "light/dark" variable or a "red/green/blue" colour variable into the state that is irrelevant for navigation. This task sequence is designed to demonstrate that if an algorithm learns to correctly generalize across different states, then the algorithm can learn to solve the maze navigation task faster than an algorithm that does not generalize correctly.

The schematic in Fig 6 illustrates how the task sequence is constructed and how additional state variables are introduced. In the bottom left of Fig 6, the task "Maze A Light-Dark" is constructed by augmenting each state of the Maze A task with a binary "light/dark" variable. As an agent transitions between different grid locations, this binary variable switches with equal probability. By adding this variable, the state space is doubled to 200 states. Note that the state $s$ will be communicated to the agent as an index that ranges from 0 to 199. The agent is not informed about the fact that states are augmented by a binary variable. To determine how this 200 state light-dark maze can be compressed, the agent would have to infer that state 0 and state 100 are equivalent and can be compressed to one latent state, for example.

The task "Maze B Coloured" (second map in bottom row of Fig 6) is constructed by augmenting the right half of the maze with a "red/green/blue" colour variable. In this case, states corresponding to the left half of the maze are not changed, but states that correspond to the

right half are augmented with either the colour red, green, or blue. Intuitively, as the agent transitions into the right half of the maze, it will observe a coloured grid cell and colours will randomly switch between either red, green, or blue. Conforming to the light-dark maze construction, the state is presented to the agent as an index ranging from 0 to 199. The agent is not given a state in a factored form, for example a grid position and colour.

The bottom row of Fig 6 depicts the five-task curriculum. In this experiment, an agent can either learn how to maximize reward in each of the 200-state tasks or learn how to compress each task into 100 latent states, generalize information across different tasks, and ultimately learn an optimal policy faster and generate higher total reward.

Fig 7 presents the results of the learning experiment conducted on the maze-task curriculum. The average-per-task episode length of each algorithm is plotted in Fig 7A. Because each task is a navigation problem, a low average episode length indicates that an algorithm reaches the rewarding goal using fewer time steps and can generate on average more reward per time step. For Q-learning, the average episode length per task remains roughly constant (blue curve

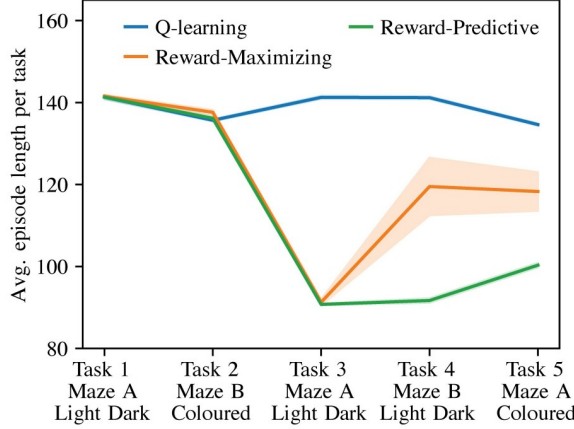

(A) Avg. episode length comparison for each learning algorithm.

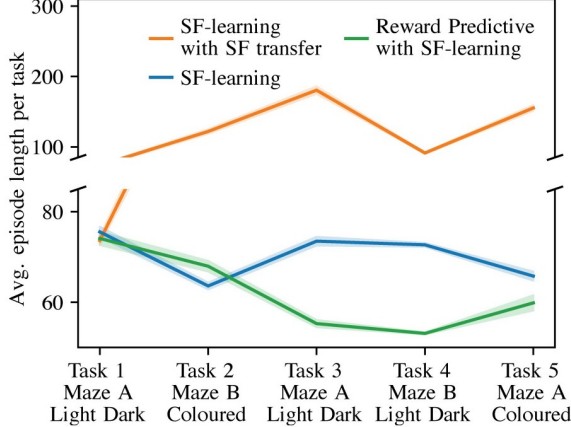

(B) Avg. episode length comparsion of transferring SFs across tasks with the reward-predictive model.

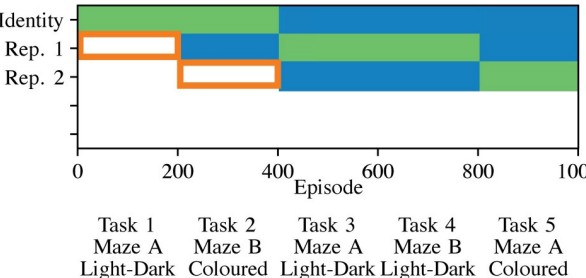

(C) Posterior probabilities during learning for the reward-predictive model.

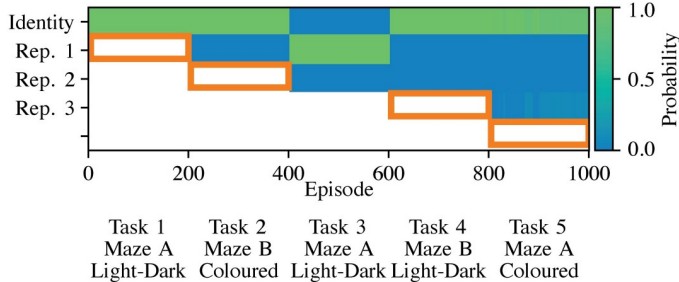

(D) Posterior probabilities during learning for the reward-maximizing model.

**Fig 7. Transferring state representations influences learning speed on the maze curriculum.** (A) Performance comparison of each learning algorithm that uses Q-learning to obtain an optimal policy. The reward-predictive model identifies two state abstractions and re-used them in tasks 3 through 5, resulting in faster learning than the reward-maximizing model. (B) Performance comparison of each learning algorithm that uses SF-learning to obtain an optimal policy. Similar to (A), the reward predictive model identifies two state abstractions and re-used them in tasks 3 through 5. Re-using previously learned SFs across tasks (orange curve) degrades performance. (A, B) Each experiment was repeated ten times and the average across all repeats was plotted. The shaded areas indicate the standard errors of measure. For each experiment, different learning rates and hyper-parameter settings were tested and the settings resulting in the lowest average episode length are plotted. Supporting S3 Text describes the tested implementation and hyper-parameters in detail. (C, D) Plot of the posterior distribution as a function of training episode. The orange rectangle indicates tasks in which the agent used the identity abstraction to learn a new state representation that was added into the belief set after 200 episodes of learning.

in Fig 7A), because Q-learning does not transfer information across tasks. In comparison, the reward-predictive non-parametric Bayesian model achieves a significantly lower average episode length on tasks three through four. This behaviour is explained by the posterior plotted in Fig 7C and 7D. On the first two tasks, the reward-predictive model adds two new state abstractions into its belief set (Fig 7C). During training on task 1 and task 2, this model uses the identity state abstraction and does not (and cannot) generalize across two different states. Consequently, there is no difference in performance between the reward-predictive model and Q-learning. Task 1 and task 2 expose the agent for the first time to a light-dark and a coloured maze and after learning in these two tasks the reward-predictive model adds a new state abstraction into its belief set (orange boxes in Fig 7C, left panel). From task 3 onward, the agent detects within the first few episodes which state abstraction to re-use in which task, resulting in faster learning and consequently shorter average episode lengths on these tasks. Specifically, on Task 3 and Task 4 the reward-predictive model re-uses the state abstraction learned in Task 1, though these tasks use different mazes (Maze A and Maze B in Fig 6); indicating that the learned state abstraction only models light/dark state equivalences and is independent of the transitions and rewards themselves. Similarly, on Task 5 the reward-predictive model re-uses the state abstraction learned in Task 2, despite both tasks using different mazes; indicating that this state abstraction only models colour state equivalences and does not depend on the transitions or rewards of either maze. These results demonstrate that the reward-predictive model is capable of extracting two state abstractions, one for the light-dark scenario and one for the coloured scenario, and re-using these state abstractions.

In contrast, the reward-maximizing model only performs comparably to the reward-predictive model on the third task (orange curve in Fig 7A). The posterior probability plot for this model (Fig 7D) indicates that only on task 3 a previously learned state abstraction is re-used. This re-use occurs because the first and third tasks are identical and the first task's solution can be repeated on the third task. For all other tasks, the reward-maximizing model introduces a new state abstraction into its belief set. This supports the hypothesis that the reward-maximizing model effectively memorizes a solution for each task and can only repeat previously learned solutions.

Fig 7B compares the average episode length of the reward-predictive model with transferring and adjusting SFs, the system used in prior work [11–14, 17, 18, 29–31]. (Supporting S3 Text provides a description and re-production of how re-using previously learned SFs leads to faster convergence.) In the tested grid-world tasks, we found that our SF-learning algorithm implementation in combination with the used initialization heuristics converges faster to an optimal policy than the Q-learning algorithm, resulting in a shorter average episode lengths. The reward-predictive model can be adopted to use the SF-learning algorithm instead of the Q-learning algorithm and this model is presented in Fig 7B. The blue curve in Fig 7B plots the average episode length when the SF-learning algorithm is used to find an optimal policy. In this simulation, the SF-learning algorithm does not transfer a representation and instead resets its weights when switching between tasks. The orange curve plots the average episode length when the SF-learning re-uses previously learned SFs instead of resetting its representation. Fig 7B demonstrates that re-using SFs degrades performance on the maze task sequence while the reward-predictive model outperforms a SF-learning baseline. On tasks 3 and 5, the reward-predictive model outperforms the SF transfer method because the reward-predictive model identifies which state abstraction to use in which task, as previously discussed. Note that when transitioning from task 3 to task 4, the underlying light-dark state abstraction is not changed whereas in all other task changes the underlying state abstraction is changed as well. This result

suggests that SFs themselves implicitly incorporate parts of the state abstraction helping the SF-learning algorithm to converge faster in task 4.

## Comparison to transferring successor features

Lastly, in this section, we illustrate the differences between re-using reward-predictive state abstractions and re-using successor features themselves [11, 12, 18]. Although reward-predictive state abstractions can be extracted from successor features, the resulting abstraction is a more abstract aspect of an MDP than simply reusing the successor features. Fig 8 presents a guitar-playing example to illustrate this idea. In this example, the task is to play a guitar scale, a sequence of notes such as C-D-E-F-G-A-B. On a guitar, the note "C" can be played by holding down a finger at one out of multiple possible locations on the fret board, as illustrated in Fig 8A. (Even within the same octave, the note "C" can be played in up to five different ways.) A skilled guitarist has internalized a representation that links fret-board positions to the notes they produce. In this example, a reward-predictive state abstraction captures this aspect of mapping all positions on the fret board to a latent state of playing the note "C".

The guitar-scale task illustrated in Fig 8A is constructed such that the agent always starts at a separate start state. To play a scale correctly, the agent has to select an action sequence that corresponds to playing the note sequence correctly. The state is represented as a bit matrix, where each entry corresponds to a position on the fret board. In the guitar-scale task the agent transitions through a sequence of fret-board locations by playing a sequence of notes. Rewards are only maximized across time if the agent plays the correct scale (Fig 8A, bottom schematic).

For a sequence of two guitar-scale tasks, Fig 8B compares the performance of a reward-predictive model with that of transferring previously learned SFs. Note that these two guitar-scale tasks differ in their transitions, rewards, and optimal policy. While all algorithms perform similarly in learning the first scale (given that they have to learn the abstraction), only the reward-predictive model (green curve) exhibits transfer to the second scale. Fig 8C plots the reward obtained in each episode for both the reward-predictive model and the SF transfer algorithm and illustrates that the reward-predictive model obtains an optimal policy faster on the second task. This performance improvement can be attributed to the fact that the reward-predictive model builds an internal representation that more closely models how to generalize across different fret-board locations, which is invariant to the scale (i.e, the reward sequence is identical if the agent correctly plays the scale in any of the octaves). Because only equivalences across fret-board locations are modelled, one would also expect a similar performance improvement for any randomly chosen scale. In contrast, SFs encode the visitation frequencies of future (latent) states under a specific policy, a property that changes between the two tasks. Note that this result is not generated because a portion of the note sequence overlaps between the two scales (C-D-E-F-G), otherwise the SF transfer algorithm would exhibit positive transfer on the second scale. Thus, the performance discrepancy in Fig 8B comes about because SFs and reward-predictive state abstractions model different aspects of an MDP.

## Discussion

In reinforcement learning, the agent's goal is to find a reward-maximizing policy. But, whereas typical RL applications pertain to a single MDP, in a lifelong learning scenario (such as that confronted by biological agents), the objective is to maximize reward across a variety of environments. For this purpose, it is critical to discover state abstractions that can be efficiently reused and transferred across a variety of situations. While several approaches exist for discovering useful state abstractions that reduce the complexity of a high dimensional task environment (e.g., using deep neural networks) in an attempt to maximize reward, this article

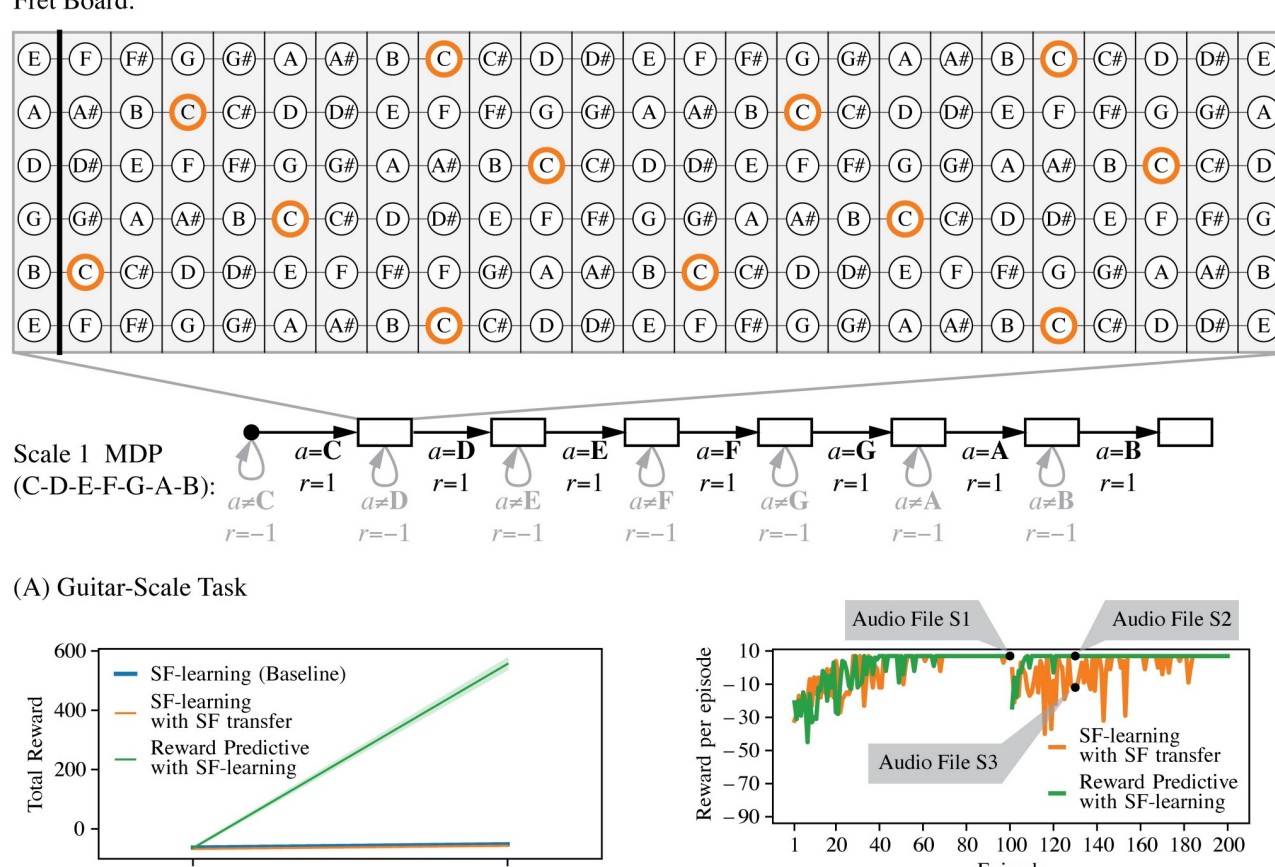

**Fig 8. Guitar-playing example.** (A): Guitar-Scale task for scale C-D-E-F-G-A-B. The fret board is translated into a bit matrix, where each entry corresponds to one circle. Because the note "C" can be played at multiple fret-board locations (orange circles), each location is mapped to the same latent state. The bottom schematic illustrates how the guitar-scale MDP is constructed for one octave: Starting at the start state (black dot), the agent progresses through different fret-board configurations by selecting which note to play next. Note that the illustrated state sequence is repeated three times, once for each octave. (The schematic illustrates only one chain to simplify the presentation.) Which octave is played is determined at random and the transition from the start state (black dot) into one of the fret boards that correspond to the note "C" is non-deterministic. This assumption allows us to reduce the action space from 60 fret board positions to 12 notes (A, A#, B, C, C#, D, D#, E, F, F#, G, G#, A). For each correct transition, a reward of zero is given, and for each incorrect transition a reward of −1 is given. The last fret board (a fret board corresponding to the note "B" in this example) is an absorbing state. (B): Total reward for each algorithm after first learning an optimal policy for Scale 1 (C-D-E-F-G-A-B) and then learning an optimal policy for Scale 2 (A-B-C-D-E-F-G). Each algorithm was simulated in each task for 100 episodes and each simulation was repeated ten times. The supporting S3 Text provides a detailed description of all hyper-parameters. (C): Reward per episode plot of one repeat for both the SF transfer and reward-predictive model. For the first 50 episodes, which are spent in scale task 1, both algorithms converge to an optimal reward level equally fast and learn to play the scale correctly. A recording of the optimal scale sequence is provided in supporting S1 Audio File. On scale task 2 (episodes 51 and onward), the reward-predictive model can re-use a previously learned state abstraction and converge to an optimal policy faster than the SF transfer algorithm. After only ten episodes in scale task 2, the reward-predictive model has learned how to play the scale correctly (please refer to supporting S2 Audio File) while the SF transfer algorithm has not yet converged to an optimal policy and does not play the scale correctly (please refer to supporting S3 Audio File).

demonstrates that, for longer term benefits, an agent should focus on learning reward-predictive state abstractions. Our findings indicate that such abstractions permit an agent to discover state spaces that can be re-used by way of analogy to previously seen state spaces, without requiring the details of the task (transitions, reward functions, or optimal policy) to be preserved.

Our initial simulations considered situations in which a single abstraction could be transferred to a subsequent MDP. However, in a lifelong learning scenario, one must consider multiple possible abstract structures that may pertain to any novel situation. When a musician picks up a banjo, they may quickly recognize its similarity to other string instruments—even those with alternate tuning—and efficiently learn to play a scale; the same musician may re-use a different structure when attempting to master the accordion. Previous theoretical work relied on non-parametric Bayesian clustering models that assess which of several previously seen structures might apply to a novel situation and be flexibly combined in a compositional fashion [3], a strategy supported by empirical studies in humans [16]. However, such an approach still requires the agent to recognize that the specific transition function and/or the reward function is portable to new situations. Here, we applied a similar non-parametric Bayesian agent to cluster reward-predictive state abstractions, affording "zero-shot" transfer of state representations to novel environments that are only similar by way of analogy to previously seen scenarios. Because the reward-predictive model can identify which state abstractions are embedded in a task and re-use these state abstraction to accelerate learning, the presented results suggest that reward-predictive state representations generalize across tasks.

Biologically, our findings motivate studies to investigate whether brain systems involved in representing state spaces, such as the hippocampus and orbitofrontal cortex [32–34], have learning rules that are guided by minimizing reward-predictive loss, rather than simply minimizing the Bellman error as in classical temporal difference learning rules leveraged by striatal dopaminergic systems [35, 36]. Indeed, dopaminergic learning signals themselves are diverse, not only conveying reward prediction errors used for optimizing actions, but with some signals (perhaps projecting to distinct circuits) appearing to be used to learn about state transitions that permit subsequent transfer [11, 18, 37, 38]. Our simulations motivate more tailored experiments to investigate the potential role of such signals in compressing state representations such that they can be analogically reused.

Existing experiments searching for neural and behavioral correlates of the SR [11, 18] have not varied both rewards and transitions, because (unlike the reward-predictive model), the SR is not robust to these changes across environments. Our work motivates the development of targeted experimental designs that would test if human subjects can reuse a latent structure that is present in a set of tasks despite variations in transitions and rewards. For example, one could design a human subject study similar to [16] where participants solve a sequence of grid-world navigation problems, but augment the design to test if subjects reuse a latent structure present in a set of tasks despite variations in transitions and rewards, similar to the task sequence presented in Fig 5. As illustrated in Fig 6, the specific pattern of generalization across tasks is predicted to vary depending on whether agents use reward-predictive state abstractions or re-use SR abstractions. Thus, our work provides a concrete testable behavioral prediction that would discriminate between our work and existing work.

Offline hippocampal replay has been proposed to reflect sampling from a model to train model-free RL and facilitate planning [39–42]. Our work provides a predicted amendment to this notion: we suggest that replay may be prioritized in such a way that facilitates the construction of reward-predictive state abstractions. In our work on learning (Learning to transfer multiple state abstractions), while the agent is first interacting in a novel MDP, it retains an identity (i.e., un-compressed) state abstraction. Only after sufficiently learning and interacting in this task, the agent can then construct a new state abstraction that can be used for planning in the future. Indeed, for efficient learning and generalization, retaining the identity map while learning is critical; otherwise the agent is likely to create a sub-optimal abstraction that will not generalize. We suggest that the online use of the identity matrix may depend on the highly pattern-separated and conjunctive representations in the hippocampus, whereas the more

abstract representations that facilitate generalization and transfer may be cortical [43]. Moreover, we speculate that one way this abstraction could be learned offline would be if, during replay, hippocampal events could be sequentially sampled from regions of the state space that are most similar in a reward-predictive sense (i.e., those that incur the least reward-predictive loss). In this way, an abstract graph-like structure suitable for future planning could be constructed [44, 45] but further augmented so that it does not depend on temporal adjacency of transitions within the graph itself, but rather in terms of the ability to predict future expected reward sequences—facilitating a deeper form of transfer. This reward-predictive loss function for guiding replay may also shed light on recent studies in rodents demonstrating that replay is biased toward recently received rewards (e.g., food) rather than those that are currently desired (e.g., water) after revaluation, even though behavior is directed toward the desired one [46]. While this pattern is counter-intuitive from the perspective that replay is used for future planning, it accords with that expected from an algorithm that compresses the state space based on reward-predictive representations, where reward is defined by the previously experienced reward function. Consequently, these representations do not generalize to any arbitrary task and are restricted to variations in transitions, rewards, and optimal policy. This restriction of reward-predictive state abstractions can be observed in Fig 1, where a representation learned for the light-dark maze would not be re-used on the coloured maze. Because the presented model demonstrates that generalization across different rewards and transitions is possible, future studies on replay would test subjects for generalization across different tasks instead of only testing for recall of a previously observed task structure.

The Tolman-Eichenbaum machine [47, 48] presents a model for generalization in the hippocampal-entorhinal system [49]. Similar to reward-predictive state abstractions, this model learns a latent representation that is predictive of future outcomes or stimuli but is also tied to a fixed transition function. While this model is not formulated in the usual RL framework, predicting future outcomes or stimuli can also be understood as a form of reward prediction. However, this model is trained directly on entire interaction sequences to predict future outcomes, and the learned representations are thus tied to the transition function. The presented transfer examples and simulations illustrate that reward-predictive state abstractions are not restricted by these limitations and can be directly re-used, assuming certain state equivalences are preserved.

Our approach also stands in contrast to prior attempts to leverage SFs [11–14, 17, 18, 29, 30] in which the SFs themselves are used to initialize learning in novel environments. Such an approach can accelerate learning in some situations, but it can be fragile to changes in the optimal policy [14] and transition function. A similar effect has also been shown for variations of reward-maximizing state abstractions [5], but these abstractions are also adjusted to each task, similar to SFs. While prior work mitigates this re-learning by associating a novel task with one out of multiple previously learned SFs [13, 31], these methods still rely on initializing learning with a previously learned representation to obtain a performance gain over solving a task from scratch. Universal successor feature approximators (USFA) [50] mitigate the dependency of previously learned SF to a single policy by defining SFs as a function $\psi^{\pi}(s, a; \boldsymbol{w})$, where the weight vector $\boldsymbol{w}$ describes a particular MDP. While this approach only requires learning one SF representation function for a family of policies, this model also assumes fixed transition functions. In contrast, reward-predictive representations have the ability to abstract away irrelevant task features and these abstractions can be re-used without re-learning them. While the presented reward-predictive model transfers state abstractions across tasks, this model has to re-learn how individual latent states are associated with one-step rewards or SFs for each task. In fact, the presented abstraction transfer models could be combined with prior work [3, 13,

16, 31] that transfers SFs, latent transition functions, or latent reward functions to integrate the benefits of each transfer system.

In related work [17], the SR of an MDP was compressed using PCA and the obtained representations were demonstrated to be suitable for transfer and connections to place cells and grid cells in the hippocampus. However, this compressed SR constructs a representation of the transition function itself, and hence transfer is again limited to environments that share the same transition function. In contrast, reward-predictive state abstractions separate the transition dynamics (and the SR) from the compression on the state space itself, and thus generate a latent state representation of a task exploiting analogical task equivalences. Latent state abstractions are not tied to particular transitions [3, 4], and can thus circumvent this dependency without adjusting the transferred representation itself.

While reward-predictive state abstractions do not limit an agent's ability to obtain an optimal policy for an MDP [9, 51], the solution space of possible reward-predictive state abstractions is far more constrained. Prior deep learning models [52] construct latent state representations as part of a model-free and model-based hybrid model that constructs a latent state representation and extracts the underlying state-transition dynamics. In contrast to their method, reward-predictive state abstractions compress the state space by generalizing across states that generate identical future expected reward sequences. While this article uses an existing SF model [9] to compute $l_{\text{predictive}}$, several other methods exist to evaluate reward-predictive state abstractions [53–55].

## Limitations and future directions

With the exception of Fig 3C, each simulation experiment assumes that a given task has an (unknown) state-abstraction embedding. In this case, there always exists a state abstraction which, if discovered, would allow any learning algorithm to find an optimal policy. A case that has not been studied in this article and is left for future work is the case when a task is over-compressed (i.e., lossy compression). Over-compressing a task induces approximation errors, because the compression removes too much information or detail from the state space such that accurate predictions are no longer possible [5, 9]. If only the latent state is given as state input to an algorithm like Q-learning, the algorithm may not converge and learn an optimal policy because the latent state is only providing partial information about the actual state of the task. One could analyze the problem as a partially observable MDP (POMDP) [56], but algorithms that can solve POMDPs also maintain a belief about which actual state they are in. In this literature, the actual state is assumed to be unknown to the agent. Because this work assumes that the actual state is known to the agent, the benefit of using such an algorithm is not clear in the case where a task is over-compressed by a state abstraction. Under what assumptions algorithms like Q-learning can be combined with state abstractions that over-compress a task is left for future work.

The presented results consider finite MDPs, allowing the algorithm to tabulate a value or latent state for each possible state. Another direction of future work is to extend the presented models and algorithms to larger state spaces, such as images. Such an extension would integrate neural networks or deep learning techniques, and allow the presented models to be applied to more complex tasks, such as computer games [22] or visual transfer tasks that can also be used in a human subject study [4].

## Conclusion

The presented results suggest that reward-predictive state abstractions generalize across tasks with different transition and reward functions, motivating the design of future transfer

algorithms. The discussed connections to predictive representations in the brain and generalization in human and animal learning motivate further experiments to investigate if biological systems learn reward-predictive representations.

## Supporting information

**S1 Text. State abstractions.**
(PDF)

**S2 Text. Successor features identify reward-predictive state abstractions.**
(PDF)

**S3 Text. Hyper-parameter selection and implementation of learning experiments.**
(PDF)

**S1 Fig. Transfer experiment adopted from [14].** (A) In this experiment each algorithm was simulated on a sequence of grid-world maps. For each grid map, the agent starts at the blue grid cell and navigates to the green goal cell to collect a reward. The transitions are the same as described in Fig 6, but these tasks do not contain any barriers. (B) Plot of the average episode length as a function of the episode for the SF-learning agent. The gray lines indicate the start or end of learning in one of the four tasks. After a certain number of episodes, the SF-learning algorithm can find an optimal policy that navigates across the map in about ten time steps. (C) Plot of the average episode length for each task and each tested algorithm. Each simulation was repeated 20 times and averages across repeats are plotted. Standard errors indicated by the shaded areas.
(TIF)

**S2 Fig. Episode length averaged across tasks two through four from S1(A) Fig.** (A) Episode length of Q-learning and SF-learning under different transfer strategies when optimistic initialization is used. The configuration "Q-Learning Q-val. Transfer" re-used previously learned Q-values. The configuration "SF-Learning SF transfer" only re-uses previously learned SFs while the configuration "SF-Learning SF and reward transfer" re-uses both SFs and the learned one-step reward predictions. (B) In this experiment both algorithms are initialized to produce zero Q-values and an $\varepsilon$-greedy exploration policy is used. This exploration strategy selects actions uniformly at random with $\varepsilon$ probability and with $1 - \varepsilon$ probability actions are selected greedily with respect to the current Q-value predictions. At the beginning of training $\varepsilon = 1$ (uniform random action selection) and by episode 80 $\varepsilon$ was decreased to zero (greedy action selection) using linear interpolation.
(TIF)

**S3 Fig. Model parameters $\alpha$ and $\beta$ control the belief space size of the non-parameteric Bayesian online learning model (Learning to transfer multiple state abstractions).** (A) Avg. episode length of the reward-maximizing model. (B) Avg. belief space size of the reward-maximizing model. (C) Avg. episode length of the reward-predictive model. (D) Avg. belief space size of the reward-predictive model.
(TIF)

**S1 Audio File. Sound version of the optimal policy in the scale task 1.**
(WAV)

**S2 Audio File. Sound version of the optimal policy in the scale task 2.**
(WAV)

**S3 Audio File. Sound version of the SF transfer algorithm's policy after learning for 25 episodes in scale task 2.**
(WAV)

## Acknowledgments

We would like to thank Rex Liu, Nicholas T. Franklin, and Alana Jaskir for insightful comments on previous drafts of this article.

## Author Contributions

**Conceptualization:** Lucas Lehnert, Michael J. Frank.

**Formal analysis:** Lucas Lehnert.

**Investigation:** Lucas Lehnert, Michael L. Littman, Michael J. Frank.

**Methodology:** Lucas Lehnert, Michael L. Littman, Michael J. Frank.

**Resources:** Michael L. Littman.

**Software:** Lucas Lehnert.

**Supervision:** Michael L. Littman, Michael J. Frank.

**Writing – original draft:** Lucas Lehnert.

**Writing – review & editing:** Lucas Lehnert, Michael L. Littman, Michael J. Frank.

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
