## [Decision Letter · Decision Letter 0]

1 Jun 2020

Dear Mr. Lehnert,

Thank you very much for submitting your manuscript "Reward-predictive representations generalize across tasks in reinforcement learning" for consideration at PLOS Computational Biology.

As with all papers reviewed by the journal, your manuscript was reviewed by members of the editorial board and by several independent reviewers. In light of the reviews (below this email), we would like to invite the resubmission of a significantly-revised version that takes into account the reviewers' comments.

The comments are lucid and I won't reiterate them here. However, I'd like to amplify a point made by one of the reviewers: the manuscript as currently written has very little connection to biology. This would not be a problem if you were submitting to a computer science journal, but this is a biology journal. In order for your manuscript to be publishable in PLOS CB, you need to engage with the relevant biology more directly. In particular, how might your proposed model be implemented in the brain? Is there any evidence for this implementation? As I'm sure you know, there has been a concerted effort to find behavioral and neural correlates of the successor representation; would your model make the same predictions or different ones? I presume different, but it's hard for me to tell exactly what the model would say about those experiments. All that being said, you don't need to dramatically change the length or structure of the paper; it would be sufficient to address these issues in the Discussion. I just want to emphasize that however you address them, you should speak directly to empirical facts about biology and not just general concepts.

We cannot make any decision about publication until we have seen the revised manuscript and your response to the reviewers' comments. Your revised manuscript is also likely to be sent to reviewers for further evaluation.

Sincerely,

Samuel J. Gershman

Deputy Editor

PLOS Computational Biology

Reviewer's Responses to Questions

**Comments to the Authors:**

Reviewer #1: Lehnert and colleagues discuss state abstractions and their role in future performance on RL tasks. Specifically, define separate state abstractions into those that support reward maximization and those that support reward predictions in current tasks. They demonstrate that reward predictive state abstractions are more beneficial to reward maximization on future tasks than reward maximizing state abstractions are. More prosaically, they suggest a good state abstractions are those that preserves the causal chain of rewards.

They first demonstrate this point by enumerating all possible state abstractions in 9-state MDPs and showing that the state abstractions that are empirically reward predictive transfer better than the reward maximizing ones.

The authors subsequently develop a Bayesian model to infer upon state abstractions for appropriate reuse between tasks. They show that the Bayesian model equipped with the reward predictivity (from the linear successor feature model (LSFM) arXiv preprint of Lehnert & Littman 2019) of each state abstraction is able to successfully infer that only two state abstractions are needed, whereas a Bayesian model equipped reward maximizing information is not.

Lastly, the authors consider learned state abstractions. Reward maximizing state abstractions are obtained from clustering Q-values. Reward predictive abstractions are learned via gradient descent on a LSFM objective function, and subsequent clustering of states. They demonstrate that, along with the above Bayesian model, reward predictive state abstractions allow for faster learning of new RL tasks that share the same underlying state space.

COMMENTS

Overall, I like this paper and think it should be published. Nevertheless, there are a few places where the impact, broader appeal and clarity of this paper could be improved upon:

1. I had some difficulty understanding the results of the second experiment. For each of the two state abstractions (\\phi_A and \\phi_B) that define the task, there are many state abstractions that are just as reward predictive as each other. For example, the identity abstraction is always as reward predictive as \\phi_A or \\phi_B. With this in mind, I don’t understand how the reward predictive Bayesian model only uses two abstractions even when \\alpha is high. I can think of two possible explanations, neither of which I could find in the paper. 1) You only consider state abstractions that compresses the 9-state MDP to a 3-state MDP. 2) The loss l_{predictive} contains more information than reward predictivity i.e. it promotes state space compression. Some clarity on these results would be helpful.

2. Following on from the second point above, the notion of reward predictive state abstractions seemed to be highly tied to LSFMs. Is it really the reward predictivity that’s driving things or LSFM? For example, if Figure 5 was repeated but using the empirical definitions of reward maximization and reward predictivity from Fig 3 would you get the same results? If LSFM is critical, then this should be mentioned and a more thorough introduction to them may be necessary.

3. The discussion on biological relevance is fairly limited and the authors do not go so far to suggest possible experiments. A more thorough discussion with behavioral and neural experiments would be a nice addition. Perhaps there are behavioral errors that distinguish reward predictive state abstraction from reward maximizing ones?

4. This wasn’t the clearest / easier-to-read paper. If for each figure the sub figures had labels A, B, C etc it would make all figures much easier to parse and refer to in the text. There are also lots of repetitions in the text e.g. Line194-197 repeats what was said 2 sentences before.

MINOR COMMENTS

Figure 7c is mislabeled

Tau/beta in Figure 5

Line 589 – state 0 and state 100?

Finally, I don’t want to make undue additional work for you guys, so feel free to say the above suggestions don’t makes sense etc.

Reviewer #2: I found the topic of this paper really interesting and exciting. I think that, in general, a formal account of what makes useful state abstractions of the type presented in this paper, will be very important for the psychology and neuroscience of human abilities to generalize experience from one task and apply it to new tasks. The objective that such representations should be able to predict arbitrary reward sequences, and that the linear successor feature models can be used to identify representations that fulfill this objective, is an interesting insight that I think will be thought provoking for future work in this area.

Major comments:

My only major comment for the paper regards the claim that the approach of transferring reward predictive state representations is well suited for situations in which the task transition structure changes between tasks. The paper makes a point here to contrast this ability with that of the successor representation/features approach which is unsuited for this situation. I’m not sure I was convinced however of the clear dominance of this approach however by any computational experiments that they presented. In particular, the authors only test this ability in families of MDPs that have a common underlying generative process, in which the difference they share a “true” state abstraction (e.g. Fig 3 b). The only experiment in which there is no true underlying abstractions is presented in (Fig 3c). However, for this experiment, only the reward function changes, but not the transition function. So, I think it might be useful to provide a simulation similar to figure 3c, but for a situation where transitions change between tasks, but there is no singular underlying generative abstraction to see how the reward predictive abstraction approach performs.

Additionally when successor features approach and the reward predictive state representation approach are compared in Fig 6, it is not clear that reward predictive representations perform better than the successor features approach, even though lots of transitions change in this task. So, it might also be useful to present some sort of task where the reward predictive state representation approach is able to perform much better than SF due to its ability to transfer state abstractions between tasks with transition changes.

Minor Comments:

Then, I have a number of minor comments mostly regarding clarity:

Section 2: I think the discussion of LSFMs being used to construct reward predictive state abstractions will be confusing to lots of readers. I found myself needing to double back and forth with the supplement section, then also read parts of the prior paper (Lehnert and Littman, 2019) to feel like I understood it. In particular, I think it’s not clear from the main text what precise role successor features play in the construction/learning of state abstractions/partitions. I’d recommend adding a bit here more fully spelling out more precisely how successor features are used to construct latent state partitions, and also providing more intuition for the central theorem in Lehnert and Littman, 2019, which relates error over reward predictions to errors in successor features. Relatedly here, in terms of understanding the use of the SF for defining reward-predictive state partitions, I felt like I didn’t understand why it was better to be able to predict the full successor representation from each latent state (feature) as opposed to just being able to predict just one-step transitions.

Section 3.2: Fig 4: In the triangle MDP plot, I think the left action from state 1 to 2 is mislabeled (a instead of c).

Fig 5: in the legend to the bar plot, should this be beta (instead of tau)?

Fig 5: For this whole figure, I think I understand this, but also found it somewhat confusing. Just for clarity, the text suggests that the main point of this figure is that when the CRP prior is completely ignored, the reward predictive agent gets the correct state abstraction, whereas the reward-maximizing agent does not. For additional parts of the figure however (which vary parameter, etc.), it looks like the approaches do not produce super clear differences in their results. Is this correct?

Fig 5: Is the identity state abstraction excluded from this analysis? Otherwise, why is it not the top scoring abstraction for both measures? Does anything about the scoring process favor state abstractions with fewer partitions?

Section 3.3 Did I understand this correctly that updates to Pr(psi|Mt,Bt,Ct) only take place offline, between tasks? I think it would be useful to present exactly at what time point each model structure (beliefs about latent states, Q values, etcl) are updated from experience. Apologies if I missed this.

Supporting information 2: For equation (1) , (2) and (3), shouldn’t P and M be exponentiated with each summed component?

Lastly, I'd would also advocate, if there is not a word limit here, that aspects of the supplement that describe the simulation protocols should be moved to a Methods section (not in a separate document from the rest of the paper) so that readers will not have to download new documents to access these sections.

Reviewer #3: uploaded as an attachment.

**Have all data underlying the figures and results presented in the manuscript been provided?**

Reviewer #1: None

Reviewer #2: No: As far as I saw, code has not yet been provided.

Reviewer #3: Yes

PLOS authors have the option to publish the peer review history of their article (what does this mean?). If published, this will include your full peer review and any attached files.

Reviewer #1: No

Reviewer #2: No

Reviewer #3: No
---

## [Decision Letter · Decision Letter 1]

7 Sep 2020

Dear Mr. Lehnert,

We are pleased to inform you that your manuscript 'Reward-predictive representations generalize across tasks in reinforcement learning' has been provisionally accepted for publication in PLOS Computational Biology.

Best regards,

Samuel J. Gershman

Deputy Editor

PLOS Computational Biology

Reviewer's Responses to Questions

**Comments to the Authors:**

Reviewer #1: The authors have satisfactorily responded to my comments. Congratulations on the paper!

Reviewer #2: I appreciate the additional work and I think this is now a strong paper.

Reviewer #3: The authors have clarified a number of points on which I was initially unclear, and I think this already interesting manuscript is substantially improved.

In particular, how abstractions are computed is clear, and how policies are generated from abstractions is also clear. I would still call this few-shot not zero shot, because although the abstractions transfer immediately (zero-shot), the policies require additional Q-learning in the new environment, just less than otherwise would be require (few-shot).

Seems like *reward predicting* as opposed to *state predicting* or *feature predicting* is not crucial to success here. Arbitrary feature predicting should be just as good – even better when reward is sparse. I understood the authors point here to be that there might be reasons to suspect that don’t have to anchor to any external feature – in fact, a feature that is still anchored to reward is sufficiently rich, which allows the problem of selecting non-reward features to be rewarded.

Nit: From a rhetorical/intuition building perspective, might be worth revisiting the gear switching perspective or using a different example. To me, this intuitively seems like a different type of transfer – realigning abstract coordinates (reversing left+right), but keeping the same policy on top of the abstractions, as opposed to relearning policy over the same abstractions.

In addressing our concerns, this long manuscript has become very long -- I think clarity will be improved when made more concise. However, I wouldn't make this stand in the way of acceptance and I assume this will be required to meet the page limit anyway.

**Have all data underlying the figures and results presented in the manuscript been provided?**

Reviewer #1: None

Reviewer #2: Yes

Reviewer #3: Yes

PLOS authors have the option to publish the peer review history of their article (what does this mean?). If published, this will include your full peer review and any attached files.

Reviewer #1: No

Reviewer #2: No

Reviewer #3: No

---

## [Editor Report · Acceptance letter]

9 Oct 2020

PCOMPBIOL-D-20-00613R1 

Reward-predictive representations generalize across tasks in reinforcement learning

Dear Dr Lehnert,

I am pleased to inform you that your manuscript has been formally accepted for publication in PLOS Computational Biology. Your manuscript is now with our production department and you will be notified of the publication date in due course.

With kind regards,

Laura Mallard
